# Orientation and Aggregation of Polymer Chains in the Straight Electrospinning Jet

**DOI:** 10.3390/ma13194295

**Published:** 2020-09-25

**Authors:** Andrey Subbotin, Valery Kulichikhin

**Affiliations:** 1A.V. Topchiev Institute of Petrochemical Synthesis, Russian Academy of Sciences, Leninskii prosp. 29, 119991 Moscow, Russia; klch@ips.ac.ru; 2A.N. Frumkin Institute of Physical Chemistry and Electrochemistry, Russian Academy of Sciences, Leninskii prosp. 31, 119071 Moscow, Russia

**Keywords:** electrospinning, jet, polymer chains, orientation, phase separation, fiber

## Abstract

The dynamics of a straight section of a jet arising during the electrospinning of a polymer solution without entanglements, and the orientation of polymer chains in the jet were explored based on the analysis of the forces balance equation and the rheological equation of the finitely extensible nonlinear elastic model. Two modes of the jet behavior were predicted. At relatively low volumetric flow rates, the straight jet has a limited length, after that, its rectilinear motion becomes impossible, while at higher flow rates, the jet always remains straightforward. It is shown that polymer chains in a jet can be strongly stretched, which leads to phase separation in a spinning solution. Aggregation of the stretched chains was also studied and the parameters of the emerging inhomogeneous structure were predicted.

## 1. Introduction

Electrospinning is an effective and versatile method of nanofiber production for diverse technological applications [1,2]. The process is based on the ability of a polymer liquid meniscus to take on a conical shape, in which the apex emits a thin straight jet when the system is subjected to high voltage. At some distance, the straight jet often experiences whipping instability and after evaporation of the solvent, is converted into a nanofiber. Variation of an applied voltage, flow rate, and physical parameters of the liquid allow for the characteristics of the jet to be controlled. The window of the stable cone-jet mode resulting in uniform fiber formation is rather narrow and prediction of the operation area is still an urgent problem.

Three sections can be distinguished in the cone-jet structure, namely, the cone-shaped meniscus, which is suspended to the nozzle electrode; the cone-jet transition zone; and thin jet (Figure 1). According to the Taylor theory, stabilization of the cone-shaped meniscus is ensured by the action of the normal electrostatic force and capillary force [3]. However, the static cone with the angle 98.6° proposed by Taylor cannot emit a jet. A number of experiments have revealed a vortex flow inside the meniscus [4,5]. This flow can be associated with the action of the tangential electric force, which is the driving force of the jet flow.

In the transition zone, there is a change in the mechanism of the electric current flow as well as in the balance of acting forces [1,6,7,8]. Analysis of the cone-jet diameter shows that the rate of flow deformation exceeded the inverse relaxation time of the polymer chains here [9,10,11]. The total electric current in the cone-jet has three main contributions: the current of the bulk ions $I_{b}$; the conductive current of the surface ions $I_{s}$; and the surface convective current due to the fluid flow $I_{c}$: $I=I_{b}+I_{s}+I_{c}$. The bulk current dominates inside the meniscus, $I≃I_{b}$, whereas in the jet, it is mainly determined by the convective current $I_{c}$ and the surface conductive current $I_{s}$. In the transition zone, $I_{b}≃I_{c}$ or $I_{b}≃I_{s}$. Following Taylor’s arguments, the density of the free electric charge $\sigma_{f}$ on the surface of the transition zone can be estimated from the balance between the capillary force γ/b where γ is the surface tension and b is its radius, and the normal electrostatic force of $F_{n}∼\sigma_{f}^{2}/\varepsilon_{0}$ ($\varepsilon_{0}$ is the permittivity of vacuum): $\sigma_{f}∼{\left(\gamma \varepsilon_{0}/b\right)}^{1/2}$ [1,3]. The electric field of strength E inside the transition zone induces the bulk current $I_{b}∼\pi b^{2}KE$ (K is the bulk conductivity of the polymer liquid) and the surface conductive current $I_{s}∼2\pi b\sigma_{f}\mu E$ (μ is the mobility of the surface ions). The flow of the liquid leads to the convective current $I_{c}∼2\pi b\sigma_{f}v$. Here $v=Q/(\pi b^{2})$ is the average flow velocity and Q is a volumetric flow rate. If the electric field inside the transition zone is defined by the cone surface charges (i.e., $E∼\sigma_{f}/\varepsilon_{0}$, and $I_{b}≃I_{c}$), then the radius should be $b=b_{1}∼{\left(Q\varepsilon_{0}/K\right)}^{1/3}$ and the electric current is $I≃2I_{b}∼{\left(\gamma KQ\right)}^{1/2}$ [6]. When the electric field in the transition zone becomes of the order of the external field $E_{0}$, $E∼E_{0}$, another regime in which the radius of the transition zone and the electric current, respectively, are given by $b=b_{2}∼{\left(\frac{Q}{KE_{0}}\right)}^{2/7}{\left(\gamma \varepsilon_{0}\right)}^{1/7}$ and $I∼{\left(\gamma \varepsilon_{0}\right)}^{2/7}{\left(KE_{0}\right)}^{3/7}Q^{4/7}$ is produced [12]. At a given $E_{0}$ and high flow rates, the radius of the cone-jet transition zone equals $b_{2}$ whereas at low flow rates, it is $b_{1}$. With a further decreasing flow rate, the convective current decreases, therefore the cone-jet transition is defined by the condition $I_{b}∼I_{s}$. The radius of the transition zone now becomes equal to $b=b_{3}∼{\left(\mu /K\right)}^{2/3}{\left(\gamma \varepsilon_{0}\right)}^{1/3}$ and the current is I∼eγμ where e is the unit charge [13]. Analysis of the electrohydrodynamic equations showed that the emitted jet in this case has a slender conical shape [14,15].

In most electrospinning modes, the electric current is mainly determined by the bulk current $I_{b}$ and by the convective current $I_{c}$, whereas the surface conductive current $I_{s}$ is relatively small. The dynamics of the jet in this case is governed by the balance between the tangential electric force on the one hand and the capillary, viscoelastic, and inertial forces on the other hand. The contribution of the normal electric force becomes insignificant. It follows from the fact that the surface charge density of the jet decreases with a decrease in the jet radius a as $\sigma_{f}∼aI/Q$. Thus, the normal electric force also decreases, $F_{n}\propto a^{2}$, whereas the Laplace pressure γ/a inside the jet increases and cannot be compensated by normal electric force anymore, and is already compensated by the tangential electric force $F_{\tau }$. In order to ensure a smooth change in the balance of the forces in the transition zone, the normal and tangential components of the electric force and the Laplace pressure should be of the same order of magnitude. We believe that for stable outflow of the jet, the viscoelastic and inertial forces cannot exceed the capillary forces in the transition zone.

The jet shape was studied in detail both theoretically [16,17,18,19,20] and experimentally using various methods [11,21,22]. Based on the analysis of the forces balance equation, it was shown that when the tangential electric force and capillary force dominate, the jet radius scales with a distance *z* as $a(z)\propto z^{-1}$. The similar scaling behavior was obtained from the balance of the tangential electric and viscous forces [12]. In the case of the viscoelastic forces, a different scaling $a(z)\propto z^{-1/2}$ was predicted [20,21]. Finally, the balance of the electric and inertial forces gives $a(z)\propto z^{-1/4}$ [16].

The dynamics of the polymer chains and the process of fiber formation during electrospinning remain poorly studied. Experiments revealed the appearance of a phase transition accompanied by the formation of string-like structures in the jet [23,24]. This behavior is consistent with the behavior of the filaments of polymer solutions under extension [25]. It has been found that threads formed by solutions of flexible-chain polymers can show exponential thinning, which is further replaced by the emergence of blistering structures and, as a result, a fiber is formed [26,27,28,29]. Exponential thinning is associated with the unfolding of polymer chains along the stretching axis [30] and the blistering mechanism can be attributed to the flow-induced phase separation under extension [31]. A polymer/solvent demixing in polymer solutions arises due to flow-induced orientation of the polymer chains, leading to a subsequent reversal of their effective interactions from repulsive to attractive. As a result, the elongated chains tend to micro-separate and form a network of fibrils compressing laterally by squeezing and releasing the solvent out to the jet surface [32,33].

In the present paper, the cone-jet mode formed by polymer solutions without entanglements was studied. Based on the numerical analysis of the forces balance equation and the FENE-P rheological equation (the FENE-P model means the finitely extensible nonlinear elastic model with approximation of A. Peterlin to obtain a closed-form constitutive equation), we explored the orientation of the polymer chains under electrospinning and the initial stage of aggregation of the stretched chains in the straight section of the electrospinning jet.

## 2. Basic Equations

Consider a polymer solution jet emitted by cone-shaped meniscus. As far as the radius of the straight jet a(z) varies slightly with the distance z, its behavior can be described using the slender body approximation [17,18,19,20]. In the stationary regime, the flow velocity inside the jet and the extension rate are presented as follows:(1)$$v_{z}=\frac{Q}{\pi a^{2}(z)},\;\;\;\dot{\varepsilon }=\frac{dv_{z}}{dz}≃-\frac{2Q}{\pi a^{3}}\frac{da}{dz}$$

It was assumed that the electrodes generate an electric field of strength $E_{0}$ directed along the *z*-axis. The following physical parameters were used: the polymer solution was characterized by its density ρ and surface tension γ. Conductivity of ions in the solution was K.

Special attention has to be paid to the constitutive equation describing the polymer solution. In order to capture the viscoelastic effects, also in a strong elongational flow, the FENE-P model of finite extensible polymer chains was engaged [34,35]. The FENE-P model contains several parameters, namely the polymer segment length $a_{s}$, the polymer chain length L, the elastic modulus G, and the relaxation time τ. The linear viscosity of the polymer solution is expressed by means of the scaling relation η≃Gτ. The polymer chain is modeled by a non-linear spring with the extension force $f=\frac{3k_{B}T}{a_{s}L}\frac{R}{1-R^{2}/L^{2}}$, where $k_{B}$ is the Boltzmann constant, T is the temperature, and R is the end-to-end distance of a polymer chain. The number of segments in the chain is $N=L/a_{s}\gg 1$. Note, the segment length is defined through the mean-square end-to-end distance: ${\langle R^{2}\rangle }_{0}=a_{s}L$. The number of segments N is related to the molecular weight of the chain M as $N=M/M_{s}$ where $M_{s}$ is the molecular weight of a segment. For semi-dilute solutions without entanglements, $\tau \propto M^{2}$ (the Rouse time) and $G≃nk_{B}T$ where n is the concentration of polymer chains.

The constitutive equation of the FENE-P model is formulated in terms of a conformations tensor A=〈RR〉. Assuming that the stress tensor Σ is mainly determined by the polymer component and the contribution of the solvent is relatively small, we have
(2)$$\Sigma =G\frac{A/R_{0}^{2}-I}{1-\mathrm{tr}A/L^{2}}$$
The conformations tensor A obeys the equation
(3)$$\tau \left[\frac{\partial A}{\partial t}+\left(v\cdot \nabla \right)A-{\left(\nabla v\right)}^{T}\cdot A-A\cdot \nabla v\right]+\frac{A-R_{0}^{2}I}{1-\mathrm{tr}A/L^{2}}=0$$
where v is the velocity field. The velocity gradients in 3D are given by
(4)$${\left(\nabla v\right)}_{ij}=\frac{\partial v_{j}}{\partial x_{i}},\;\;\;{\left(\nabla v\right)}_{ij}^{T}=\frac{\partial v_{i}}{\partial x_{j}},i,j=1,2,3$$
and $R_{0}^{2}=\frac{1}{3}{\langle R^{2}\rangle }_{0}$.

Let us employ the cylindrical system of coordinates (z,r,φ) and assume that the jet is axially symmetric. The longitudinal electric field $E_{z}$ acting on the jet depends on z, but not on r. Below, we consider the stationary straight segment of the jet, for which it is assumed $E_{z}≃E_{0}$. This approximation does not affect the predictions of the jet behavior. The electric current passing through the jet is given by
(5)$$I≃I_{c}+I_{b}≃2\pi a\sigma_{f}v_{z}+\pi a^{2}KE_{z}$$
where $\sigma_{f}$ is the surface density of the free charges. Here, we omitted the surface current $I_{s}$, which is relatively small. The radius of the jet is estimated from the force balance equation [13]
(6)$$a^{2}\frac{d}{dz}\left(\gamma C-F_{n}\right)+\frac{d}{dz}\left[a^{2}\left(\rho v_{z}^{2}+\Sigma_{nn}-\Sigma_{zz}\right)\right]=2aF_{z}$$
where $\Sigma_{nn}≃\Sigma_{rr}$ is the normal to the free surface component of the stress tensor and C is the total curvature of the surface. As has been noted above, the contribution of the normal electric force $F_{n}$ to the overall balance of forces in the jet is relatively small; therefore we will neglect this in the future. The tangential component of the electric force is expressed via the surface charge density $\sigma_{f}$ as
(7)$$F_{z}=\sigma_{f}E_{z}$$

In our consideration, we will assume that the electric current is known, therefore the surface charge density can be expressed through the current based on Equation (5). Taking into account that the capillary term is approximately $\gamma \frac{dC}{dz}≃-\frac{\gamma }{a^{2}}\frac{da}{dz}$ and using Equation (2), the momentum Equation (6) is written as
(8)$$\frac{da}{dz}\left(\gamma +\frac{2\rho Q^{2}}{\pi^{2}a^{3}}\right)+G\frac{d}{dz}\left[\frac{a^{2}\left(A_{zz}-A_{rr}\right)/R_{0}^{2}}{1-\left(A_{zz}+2A_{rr}\right)/L^{2}}\right]=-\frac{E_{0}a^{2}}{Q}\left(I-\pi a^{2}KE_{0}\right)$$

We can dimensionalize Equation (8) and rheological Equation (3) by using the following substitutions
(9)$$a=b^{*}\bar{a},z=\ell \bar{z},A=\bar{A}R_{0}^{2}$$
where $b^{*}=\gamma /G$ and $\ell =\frac{Q\tau }{\pi b^{*2}}=\frac{Q\eta G}{\pi \gamma^{2}}$ are the intrinsic length-scales. After some transformations, Equations (3) and (8) take the dimensionless form
(10)$$\frac{d{\bar{A}}_{zz}}{d\bar{z}}+\frac{4{\bar{A}}_{zz}}{\bar{a}}\frac{d\bar{a}}{d\bar{z}}+\frac{{\bar{a}}^{2}\left({\bar{A}}_{zz}-1\right)}{1-\delta \left({\bar{A}}_{zz}+2{\bar{A}}_{rr}\right)}=0$$
(11)$$\frac{d{\bar{A}}_{rr}}{d\bar{z}}-\frac{2{\bar{A}}_{rr}}{\bar{a}}\frac{d\bar{a}}{d\bar{z}}+\frac{{\bar{a}}^{2}\left({\bar{A}}_{rr}-1\right)}{1-\delta \left({\bar{A}}_{zz}+2{\bar{A}}_{rr}\right)}=0$$
(12)$$\frac{d\bar{a}}{d\bar{z}}\left[1+\frac{\omega }{{\bar{a}}^{3}}-\frac{2\bar{a}\left({\bar{A}}_{zz}+2{\bar{A}}_{rr}\right)}{1-\delta \left({\bar{A}}_{zz}+2{\bar{A}}_{rr}\right)}-\frac{4\delta \bar{a}{\left({\bar{A}}_{zz}-{\bar{A}}_{rr}\right)}^{2}}{{\left(1-\delta \left({\bar{A}}_{zz}+2{\bar{A}}_{rr}\right)\right)}^{2}}\right]=\frac{{\bar{a}}^{4}\left({\bar{A}}_{zz}-{\bar{A}}_{rr}\right)\left(1-3\delta \right)}{{\left(1-\delta \left({\bar{A}}_{zz}+2{\bar{A}}_{rr}\right)\right)}^{3}}-\chi {\bar{a}}^{2}\left(1-\alpha {\bar{a}}^{2}\right)$$

In the above formulas, $\delta =\frac{R_{0}^{2}}{L^{2}}=\frac{1}{3N}$, $\chi =\frac{\eta IE_{0}}{\pi \gamma^{2}}$, $\omega =\frac{2\rho Q^{2}}{\pi^{2}\gamma b^{*3}}=\frac{2\rho Q^{2}G^{3}}{\pi^{2}\gamma^{4}}$, and $\alpha =\pi b^{*2}KE_{0}/I$. The total electric current is equal to twice the bulk current in the transition zone, $I=2I_{b}=2\pi b^{2}KE_{0}$, therefore $\alpha =b^{*2}/(2b^{2})$. The parameter χ depends on the strength of the electric field and also on the flow rate as far as the electric current generally depends on Q. The parameter ω depends on the flow rate and can be used to determine the magnitude of the inertia force. In our numerical calculations, we estimated χ and ω based on the values of the process parameters, which usually vary over a wide range.

The initial conditions for Equations (10)–(12) are formulated in the transition zone. It is assumed that $b^{*}$ exceeds the radius of the transition zone b. This allows the capillary forces to compensate for the large normal stresses that occur in the transition zone. The initial radius of the jet is chosen from equality of the bulk and convective currents ($I_{b}=I_{c}$): $\bar{a}(0)=1/\sqrt{2\alpha }$. In the numerical calculations, we put α=10 and α=400. The second case corresponds to a dilute solution of very long chains having a small elastic modulus G≤10 Pa [10,19]. Finally, the initial conformations of the polymer chains were assumed to be close to Gaussian (i.e., ${\bar{A}}_{zz}(0)={\bar{A}}_{rr}(0)=1$).

## 3. Orientation of Polymer Chains

Equations (10)–(12) allow the conformations tensor of the stretched polymer chains and their orientation to be found as a function of position z along the jet. We will characterize the orientation by the order parameter $s=\sqrt{\left(A_{zz}-R_{0}^{2}\right)}/L=\sqrt{\delta \left({\bar{A}}_{zz}-1\right)}$, where 0≤s≤1. For highly elongated chains, ${\bar{A}}_{zz}\gg 1$, the order parameter equals to $s=R_{z}/L$ where $R_{z}$ is the size of the chain along the jet axis. In our numerical analysis of Equations (10)–(12), we used the following three combinations of the parameters N, χ, and α: (N=200; χ=1.4; α=10), (N=2000; χ=10; α=10), and (N=10,000; χ=0.5; α=400). The values of the parameter ω were chosen in such a way as to identify different modes of the jet behavior.

The calculations showed two modes of the jet behavior. In the first mode, when the parameter ω was less than some critical value $\omega^{*}$, $\omega <\omega^{*}$, a straight jet has a finite length (i.e., it terminates at a finite distance from the transition zone). The corresponding dependences $\bar{a}(\bar{z})$, $s(\bar{z})$, and ${\bar{A}}_{rr}(\bar{z})$ are shown in Figure 2a–d. The finite jet length is related to the divergence of the derivative $\frac{d\bar{a}}{d\bar{z}}\to -\infty$ at some distance from the transition zone. This behavior arises due to the finite extensibility of the polymer chains and elastic deformation of the solution. Obviously, the slender body approximation cannot be applied at the terminal point. The order parameter monotonically increases along the jet axis to some maximum value at the end of the straight jet, whereas the transversal size of the elongated chain, which is characterized by the component ${\bar{A}}_{rr}(\bar{z})$, shows more complex behavior. It either decreases monotonously or first decreases and then increases. The jet behavior after the straight section requires special consideration. We expected the appearance of whipping instability in this case.

In the second mode, the jet always remains straightforward. The order parameter first increases along the jet axis, and after reaching the maximum value, it begins to decrease (Figure 3a–c). In the case of very long chains, N=10,000, we did not reach the maximum in our calculations (Figure 2c). We believe that this is due to the very long relaxation time of such chains. The maximum should appear at longer distances. The component $A_{rr}$ of the conformations tensor first decreases, and then starts to grow. Such behavior indicates the occurrence of the chain relaxation process. It is interesting to note that the relaxation of the transversal size of the chains begins earlier than the longitudinal one. The second mode occurs when $\omega >\omega^{*}$.

In order to elucidate the results, we plotted ${\bar{A}}_{zz}^{1/4}$ as a function of $1/\bar{a}$ (Figure 4a–c), and the Weissenberg number $\mathrm{Wi}=\dot{\varepsilon }\tau =\frac{2}{{\bar{a}}^{3}}\left|\frac{d\bar{a}}{d\bar{z}}\right|$ as a function of $\bar{z}$ (Figure 5a–c). The information in Figure 4a–c allows us to estimate the role of elasticity in the behavior of the jet. Based on the graphs in Figure 4a–c, we find component ${\bar{A}}_{zz}$ as a function of the jet radius: ${\bar{A}}_{zz}≃{\bar{a}}_{0}^{4}/{\bar{a}}^{4}$. This dependence is associated with the almost elastic deformation of the polymer solution after leaving the transition zone. If we consider a small section of the jet of length $\Delta {\bar{z}}_{0}$ and a volume $\pi {\bar{a}}_{0}^{2}\Delta {\bar{z}}_{0}$ inside the transition zone, then due to conservation of the volume, its length after displacement by a certain distance is $\Delta \bar{z}=\frac{{\bar{a}}_{0}^{2}}{{\bar{a}}^{2}}\Delta {\bar{z}}_{0}$ where $\bar{a}$ is the current radius. Thus, the longitudinal size of the chains increases ${\bar{a}}_{0}^{2}/{\bar{a}}^{2}$ times [9]. The order parameter changes according to the law $s≃{\bar{a}}_{0}^{2}\sqrt{\delta }/{\bar{a}}^{2}$. It is interesting to note that in this regime, the viscoelasticity partially compensates the inertia. This conclusion follows from the analysis of the left-hand side of Equation (12) (the second and the third terms in the brackets behave as $1/{\bar{a}}^{3}$ but with the opposite sign).

The calculations show that the Weissenberg number was large in the transition zone, Wi≫1, which is in agreement with the experiment [9,10,11]. For N=200 and N=2000 (Figure 5a,b), the Weissenberg number initially decreases with the distance when moving away from the transition zone. This decay is associated with the effect of the inertial force, the value of which in the transition zone is close to the value of the capillary force. Indeed, the inertial force leads to a slight decrease in the radius of the jet with respect to distance, $a\propto z^{-1/4}$ [16]; therefore, the Weissenberg number decreases with a decrease in the deformation rate as $\mathrm{Wi}=\dot{\varepsilon }\tau \propto z^{-1/2}$. In the first mode (the red lines in Figure 5a–c), at some distance from the transition zone, the fall gives way to rapid growth of the Weissenberg number to infinity (the derivative $\frac{d\bar{a}}{d\bar{z}}\to -\infty$), so the straight jet can no longer exist. This behavior is associated with the increasing role of the finite extensibility of the chains. In the second mode (the green lines), the Weissenberg number decreases monotonously and the chain conformations gradually relax to equilibrium.

In the case of very long polymer chains, N=10,000 (Figure 5c), the behavior is more complex. The Weissenberg number initially increases with a distance. This means that the inertial forces are relatively small in the transition zone and the jet dynamics are governed by the capillary, viscoelastic, and electric forces. Note, the capillary and electrical forces lead to an increase in the rate of stretching, $\dot{\varepsilon }\propto z$. A rapid increase in the Weissenberg number up to infinity proceeds when $\omega =2\times 10^{-6}$. Another scenario arises when $\omega =4\times 10^{-6}$: the Weissenberg number passes through a maximum and then decreases with distance. After decreasing, it begins to grow again and rapidly goes to infinity. For large values ω (we considered $\omega =2\times 10^{-5}$), the Weissenberg number decreases gradually with distance and becomes less than one. The polymer chains relax and inertia dominates with respect to the capillary and viscoelastic forces.

## 4. Dynamics of Chain Aggregation

As shown in earlier studies [31,32,33], the flow-induced orientation of the polymer chains reduces their steric repulsion and shifts their interactions from repulsive to attractive. The attraction between the polymer segments weakly depends on their orientation [36]. This leads to phase separation of the polymer solution into a polymer and solvent. In the previous section, we showed that macromolecules in the electrospinning jet become strongly stretched, therefore, phase separation can also be in this case. Below, we study this effect in more detail. For analysis of the polymer/solvent demixing, we considered a semiflexible chain of diameter d, Kuhn segment length l, and total length L so that d≪l≪L. We associated this chain model with the previous chain model that was used in the FENE-P rheological equation, assuming that the size of the polymer coils in both models was equal (i.e., ${\langle R^{2}\rangle }_{0}≃lL=a_{s}L=a_{s}^{2}N$). From here, we get $a_{s}=l$. The free energy of interactions between the segments in the third virial approximation is given by $F_{\mathrm{int}}=\frac{1}{2}B_{2}c^{2}+\frac{1}{3}B_{3}c^{3}$, where c is the concentration of the polymer segments (c=nN), and $B_{2}$, $B_{3}$ are the second and the third virial coefficients, respectively. Within standard approximation, the second virial coefficient includes the contribution from the steric repulsion and van der Waals attraction (i.e.,$B_{2}≃\frac{\pi }{2}l^{2}dk$, where $k=I\left(s\right)-\frac{\Theta }{T}$ and Θ is Θ−temperatute). The function $I(s)=\frac{4}{\pi }\langle \sin\beta \rangle$ takes into account the anisotropy of steric repulsion. Here, β is the angle between two segments and averaging is performed over the orientations of all pairs of segments. The graph of the function I(s) is shown in Figure 6. The third virial coefficient is $B_{3}≃\frac{3\pi^{2}}{32}l^{3}d^{3}I(s)$ [31].

The steric repulsion between the extended chains decreases with an increase in their orientation, therefore the balance between repulsive and attractive interactions is shifted toward attractions as the Weissenberg number $\mathrm{Wi}=\dot{\varepsilon }\tau$ increases. The polymer/solvent phase separation occurs once the osmotic modulus $\kappa =\frac{1}{k_{B}T}\frac{\partial \Pi }{\partial c}$, where Π, the osmotic pressure, becomes negative. The osmotic pressure involves the ideal gas term and contribution due to interactions,
(13)$$\Pi =k_{B}T\left(\frac{c}{N}+\frac{1}{2}B_{2}c^{2}+\frac{2}{3}B_{3}c^{3}\right)$$

We assumed that the chains were long, N≫1, therefore the osmotic modulus of the polymer solution can be written as
(14)$$\kappa ≃\frac{\pi l}{2d}\phi \left(k+3I(s)\phi \right)≃\frac{\pi l}{2d}\phi k$$

Here, $\phi =\frac{\pi }{4}d^{2}lc<\frac{d}{l}$ is the volume fraction of the polymer in the solution, which is below the transition point to the nematic phase. When the parameter k becomes negative, the polymer solution starts to separate into polymer-rich and solvent phases. The volume fraction of the polymer in the polymer-rich phase is obtained from the equality of osmotic pressure to zero: $\phi_{c}≃\left|k\right|/I(s)≃T\left|k\right|/\Theta \ll \phi$ (it is assumed that |k|≪1) [31,32,33].

Next, we will consider the kinetics of phase separation in the jet, which occurs according to the scenario of spinodal decomposition. The transitions between the states of the polymer solution in the electrospinning process are shown in Figure 7. Let us assume that the critical condition for the onset of phase separation occurs at some point $\bar{z}={\bar{z}}^{*}$ when the Weissenberg number is $\mathrm{Wi}({\bar{z}}^{*})={\mathrm{Wi}}^{*}\gg 1$ and the order parameter is $s^{*}=s({\bar{z}}^{*})$ ($k(s^{*})=0$). The spinodal decomposition of the polymer solution should occur over a certain section of length $\Delta \bar{z}=\bar{z}-{\bar{z}}^{*}$ from the critical point. In this case, the order parameter slightly increases by the value
(15)$$\Delta s=s-s^{*}≃{\bar{a}}_{o}^{2}\sqrt{\delta }\left(1/{{\bar{a}}^{2}|}_{{\bar{z}}^{*}+\Delta \bar{z}}-1/{{\bar{a}}^{2}|}_{{\bar{z}}^{*}}\right)≃s^{*}{\bar{a}}^{*2}{\mathrm{Wi}}^{*}\Delta \bar{z}$$

Here, ${\bar{a}}^{*}=\bar{a}({\bar{z}}^{*})$ and the condition ${\bar{a}}^{*}\ll \Delta \bar{z}\ll {\bar{a}}^{*}{\left|\frac{d\bar{a}}{d\bar{z}}\right|}_{{\bar{z}}^{*}}^{-1}=2{\left({\bar{a}}^{*2}{\mathrm{Wi}}^{*}\right)}^{-1}$ must be fulfilled. The last inequality gives a limitation on the polymer concentration in the solution that will be formulated below. We determine the decomposition time $t^{*}$ as $t^{*}=\ell \Delta \bar{z}/v_{z}({\bar{z}}^{*})=\tau {\bar{a}}^{*2}\Delta \bar{z}$. Obviously, inequality $t^{*}/\tau \ll 1$ is valid. To estimate $t^{*}$, consider the dynamics of concentration fluctuations δc(q,t) (q is the wave vector, t is time), which in the linear approximation obeys the equation [32,33]
(16)$$\delta c\left(q,t\right)=\delta c\left(q,0\right)e^{\Gamma (q)t}$$

Here, the growth rate Γ(q) is given by
(17)$$\Gamma (q)≃\frac{q^{2}l^{2}}{\tau_{0}}\left(\left|\kappa \right|-\frac{q^{2}l^{2}}{2}-\frac{q_{z}^{2}s^{*2}}{2q^{2}}\right)≃\frac{q^{2}l^{2}}{\tau_{0}}\left(\frac{\phi l}{d}\theta \Delta \bar{z}-\frac{q^{2}l^{2}}{2}-\frac{q_{z}^{2}s^{*2}}{2q^{2}}\right)$$
where $\theta =\frac{\pi }{2}s^{*}{\bar{a}}^{*2}{\mathrm{Wi}}^{*}\left|I^{\prime }(s^{*})\right|$ and $\tau_{0}=\tau /N^{2}$ is the segment relaxation time (we used approximation $\left|k(s^{*}+\Delta s)\right|≃\left|I^{\prime }(s^{*})\right|\Delta s$). Depending on the wave vector q, some fluctuations will grow and some will fade. Fluctuations will grow when Γ(q)>0. The maximum growth rate $\Gamma^{*}$ corresponds to the wave vector $q^{*}$, the modulus of which is $\left|q^{*}\right|=q^{*}={\left(\frac{\phi \theta \Delta \bar{z}}{ld}\right)}^{1/2}$, and the projection onto the extension axis is zero, $q_{z}=0$: $\Gamma^{*}≃\frac{1}{2\tau_{0}}{\left(\frac{\phi l\theta \Delta \bar{z}}{d}\right)}^{2}$. As a result, in the jet cross-section (the plane (r,φ)), the solution decomposes into polymer-rich and polymer-depleted regions (domains) so that the difference in concentration between the regions is δc∼c (Figure 7c).

The domains are characterized by a transversal size $\xi =2\pi /q^{*}$ and the number of chains passing through the cross-section of the polymer-rich domain is estimated as $n_{d}∼\xi^{2}cls^{*}$. Taking into account that spinodal decomposition occurs in time $t∼t^{*}$ and using the ratio $\Gamma^{*}t^{*}=\Gamma^{*}\tau {\bar{a}}^{*2}\Delta \bar{z}∼1$, we find a characteristic interval $\Delta z=\ell \Delta \bar{z}$, the characteristic time $t^{*}$, and the size of the domains ξ in the emerging inhomogeneous structure:(18)$$\Delta z≃\frac{Q\tau }{\pi a^{*2}}{\left(\frac{{\bar{a}}^{*2}d}{\phi L\theta }\right)}^{2/3},t^{*}≃\tau {\bar{a}}^{*2}\Delta \bar{z}≃\tau {\left(\frac{{\bar{a}}^{*2}d}{\phi L\theta }\right)}^{2/3},\xi ≃2\pi {\left(Ll\right)}^{1/2}{\left(\frac{{\bar{a}}^{*2}d}{\phi L\theta }\right)}^{1/6}$$

The domains are elongated along the axis *z* and their typical longitudinal size is $\xi_{z}=2\pi /q_{z}^{*}$ where the wave number $q_{z}^{*}$ is found from the function Γ(q) of Equation (17) and is equals to $q_{z}^{*}∼q^{*2}l/s^{*}$. Therefore,
(19)$$\xi_{z}≃2\pi s^{*}L{\left(\frac{{\bar{a}}^{*2}d}{\phi L\theta }\right)}^{1/3}$$

Thus, an anisotropic domain microstructure ($\xi \ll \xi_{z}\ll L$) is formed (Figure 7c). Restrictions on the interval $\Delta \bar{z}$ result in the following restrictions on the polymer concentration: $\frac{d}{Nl}\frac{{\mathrm{Wi}}^{*1/2}}{\left|I^{\prime }(s^{*})\right|}\ll \phi <\frac{d}{l}$. The last inequality is always valid when N is large enough.

Since the polymer concentration in the polymer-rich domains is less than the equilibrium concentration, $\phi \ll \phi_{c}$, in the next stage of phase separation, these domains collapse. As a result, a network of highly-oriented fibrils with diameter of $d_{f}∼\xi \sqrt{\phi /\phi_{c}}$ emerges (the number of chains in the domain remains fixed) (Figure 7d). The characteristic collapse time is of the order of $t^{*}$ and is much shorter than τ [32], therefore the network of fibrils could be fully developed in the straight jet during the stretching regime. The latest stages that include aggregation of fibrils in bundles (Figure 7e), and collapse of the network accompanied by the solvent release were studied for threads of polymer solutions in [32,33]. Since the travel time of the chains inside the jet during electrospinning is very short, these stages may not have time to be completed before the jet attains the collector. Rapid solvent evaporation can also affect the kinetics of the phase transition. All of these issues require further analysis.

## 5. Conclusions

The aim of the present theoretical work was to study the orientation of polymer chains during electrospinning of semi-diluted polymer solutions without entanglements as well as to understand how the orientation of the chains affects the dynamics of the jet and formation of the fiber. Evolution of the chain conformations along the jet was examined based on a numerical solution of the forces balance equation and the FENE-P rheological equation, allowing us to understand how the orientation of the chains affects the dynamics of the jet and formation of the fiber. This made it possible to clarify the role of the nonlinear viscoelasticity associated with the finite extensibility of polymer chains and its correlation with the capillary and inertia forces. A non-equilibrium thermodynamic approach was used to explore the kinetics of the oriented chain aggregation.

A new type of the jet instability, which differs from the Rayleigh instability and electrically driven axisymmetric and non-axisymmetric (whipping) instability due to repulsion between charges on the jet surface [1,37,38] was discovered. At a fixed electric field strength, this instability arises when the Weissenberg number is Wi>1, and the polymer solution mainly deforms elastically. In this mode, the polymer chains are stretched monotonously up to some maximum value $R_{z,\max}$, after which their further stretching becomes impossible and the jet cannot be straight anymore. The orientational order parameter, which is defined as the ratio of longitudinal size of the chain to its contour length, increases along the jet axis as $s≃{\bar{a}}_{0}^{2}\sqrt{\delta }/{\bar{a}}^{2}\propto {\left(z/M\right)}^{1/2}$ and attains the maximum value $s_{\max}=R_{z,\max}/L$ in the end of the straight part of the jet. The length of the straight jet section increases with the molecular weight linearly, $L_{j}\propto M$. We believe that furthermore, the jet exhibits whipping motion and the polymer chains relax. For larger flow rates exceeding some critical value the jet dynamics is stabilized by the inertial force, so it is always rectilinear. In this mode, the order parameter changes non-monotonically along the jet: first, it increases up to a certain maximum value (at that the Weissenberg number reduces to Wi∼1) and then decreases. The decrease is associated with relaxation of the polymer chains and is characterized by the Rouse time $\tau \propto M^{2}$. Thus, the dynamics of growth and relaxation of the order parameter slows down with an increase in the molecular weight of the chains. Transition between straightforward and whipping jet has been observed experimentally in the electrospinning of concentrated solutions of polystyrene in dimethylformamide [39].

The formation of fibers from the jet occurs as a result of the chain orientation, aggregation, and evaporation of the solvent. The structure and properties of the fiber depend on the orientation of the chains [40]. We showed that polymer chains can become strongly stretched along the flow in the straight jet section, so that their orientational order parameter reaches the value of s>0.5. The high orientation of polymer chains can lead to phase separation of the polymer solution [33]. In our model, the attraction between the chains is due to the van der Waals interaction, while the repulsion occurs due to the steric interaction that is reduced with increasing orientation of the chains. Analysis of the phase separation kinetics shows the formation of an inhomogeneous concentration pattern on the jet section of length $\Delta z∼\frac{Q\tau }{a^{*2}}{\left(\frac{d}{\phi L}\right)}^{2/3}\propto M^{4/3}$ after passing the critical point. This pattern emerged during the time $t^{*}∼\tau {\left(\frac{d}{\phi L}\right)}^{2/3}\propto M^{4/3}$, which is much shorter than the chain relaxation time τ, and is characterized by the lateral length $\xi ∼{\left(Ll\right)}^{1/2}{\left(\frac{d}{\phi L}\right)}^{1/6}\propto M^{1/3}$, and longitudinal correlation length $\xi_{z}∼s^{*}L{\left(\frac{d}{\phi L}\right)}^{1/3}\propto M^{2/3}$ (see Figure 7c). After aggregation of the chains in the concentrated domains, the inhomogeneous structure transformed into a network of fibrils, from which string-like structures are formed [32,33]. The emergence of the string-like structures in the straight jet of concentrated polymer solutions has been predicted experimentally [23,24], and this behavior correlated with our results. However, the question of the effect of entanglement remains open.

## Figures and Tables

**Figure 1 materials-13-04295-f001:**
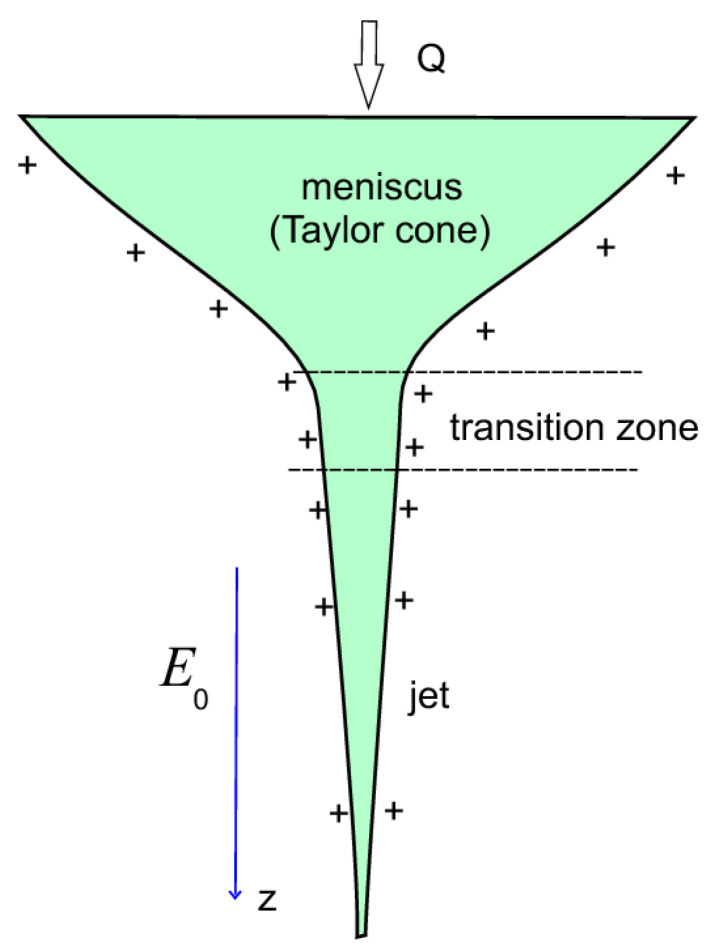
Schematic drawing of the cone-jet structure during electrospinning.

**Figure 2 materials-13-04295-f002:**
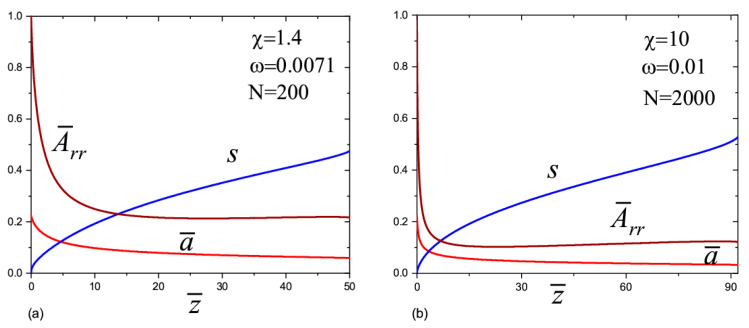
The jet radius $\bar{a}$, the order parameter s, and the component ${\bar{A}}_{rr}$ of the conformations tensor as a functions of distance $\bar{z}$ in the case when the straight jet has finite length. (**a**) N=200; χ=1.4; ω=0.0071; α=10; (**b**) N=2000; χ=10; ω=0.01; α=10; (**c**) $N=10,000;\;\chi =0.5;\;\omega =2\times 10^{-6};\;\alpha =400$ and (**d**) $N=10,000;\;\chi =0.5;\;\omega =4\times 10^{-6};\;\alpha =400$.

**Figure 3 materials-13-04295-f003:**
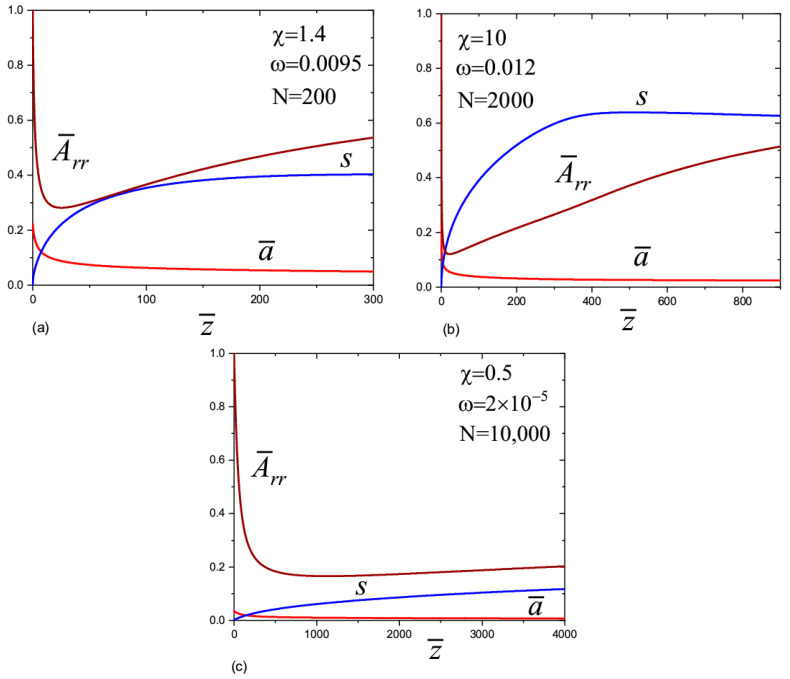
The jet radius $\bar{a}$, the order parameter s, and the component ${\bar{A}}_{rr}$ of the conformations tensor as a functions of distance $\bar{z}$ for straightforward jet mode. (**a**) N=200; χ=1.4; ω=0.0095; α=10; (**b**) N=2000; χ=10; ω=0.012; α=10; (**c**) $N=10,000;\;\chi =0.5;\;\omega =2\times 10^{-5};\;\alpha =400$.

**Figure 4 materials-13-04295-f004:**
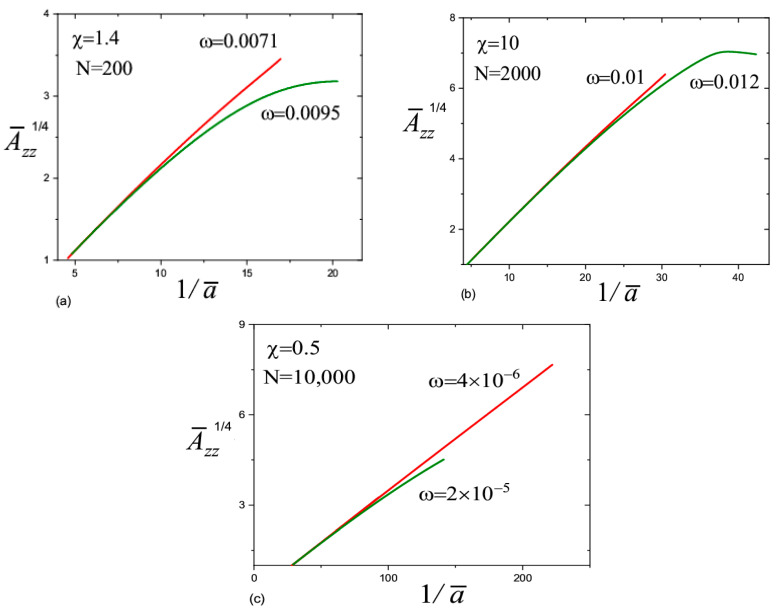
The dependences of ${\bar{A}}_{zz}^{1/4}$ vs. $1/\bar{a}$ for different parameter values. The red lines correspond to the first mode and the green lines correspond to the second mode. (**a**) N = 200; (**b**) N = 2000; (**c**) N = 10,000.

**Figure 5 materials-13-04295-f005:**
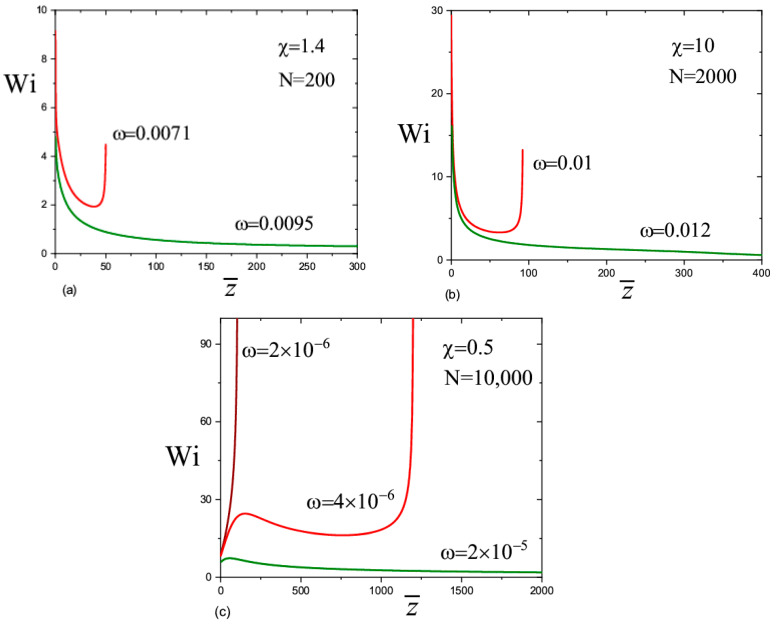
Variation of the Weissenberg number along the jet axis. The parameter values are in the figures. (**a**) N = 200; (**b**) N = 2000; (**c**) N = 10,000.

**Figure 6 materials-13-04295-f006:**
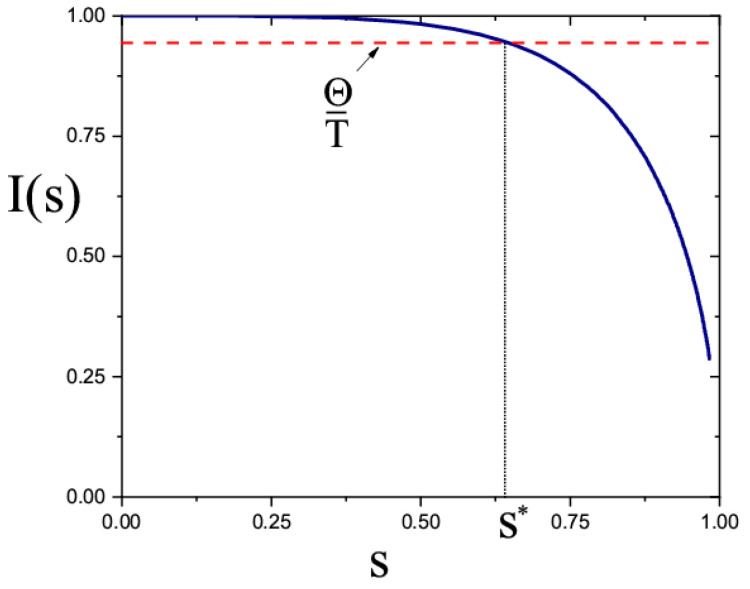
The graph of the function I(s). Dashed line implies Θ/T=const and $s^{*}$ is the critical value of the order parameter: $k\left(s^{*}\right)=I\left(s^{*}\right)-\Theta /T=0$.

**Figure 7 materials-13-04295-f007:**
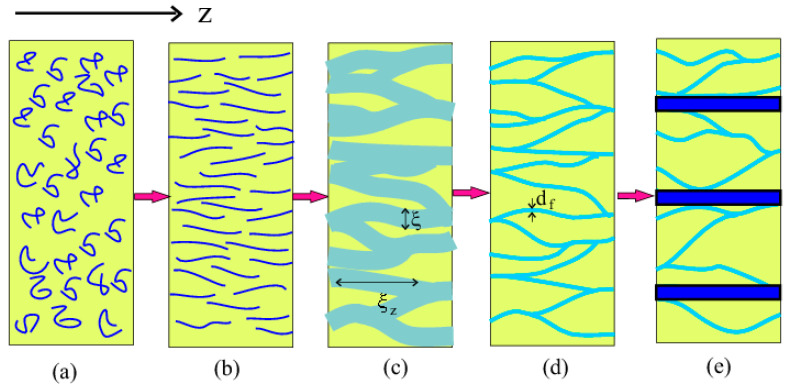
Illustration of changes in the jet with a distance: (**a**) coils of the polymer chains at the beginning of the transition zone, (**b**) stretched chains away from the transition zone, (**c**) inhomogeneous domain structure formed after spinodal decomposition, (**d**) fibril network consisting of aggregates of the oriented chains, (**e**) bundles formed by the fibrils.

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
