# Peer review of "Orientation and Aggregation of Polymer Chains in the Straight Electrospinning Jet"

_materials, 2020, doi:10.3390/ma13194295_

Round 1

Reviewer 1 Report

  • Well described methodology of an analysis of the electrospinning jet process
  • Jet stability issues are dealt with in detail
  • The predicted phase separation is a very interesting result

Author Response

Response to Reviewer 1 Comments

  • Well described methodology of an analysis of the electrospinning jet process
  • Jet stability issues are dealt with in detail
  • The predicted phase separation is a very interesting result

We sincerely appreciate the reviewer for a positive assessment of our paper.

Reviewer 2 Report

Dear Authors,

Your paper entitled "Orientation and aggregation of polymer chains in the straight electrospinning jet" provides exciting insights into the electrospinning process, and especially the jet formation stage and polymer chains stretching. I find the paper as a whole well-designed and vital for the field. The only thing I would like to suggest is to send the manuscript for English proofreading. There are several misused words and confusing statements that need to be checked by native English speakers.

Hence, I put a minor revision as my decision to give you some time to improve the language in your manuscript.

Best regards!

Author Response

Response to Reviewer 2 Comments

Your paper entitled "Orientation and aggregation of polymer chains in the straight electrospinning jet" provides exciting insights into the electrospinning process, and especially the jet formation stage and polymer chains stretching. I find the paper as a whole well-designed and vital for the field. The only thing I would like to suggest is to send the manuscript for English proofreading. There are several misused words and confusing statements that need to be checked by native English speakers.

Hence, I put a minor revision as my decision to give you some time to improve the language in your manuscript.

We sincerely appreciate the reviewer for a positive assessment of our paper. We improved English.

Reviewer 3 Report

The manuscript deals with mathematics of polymer chain behavior. However, it is important to mention experimental factors such as the influence of molecular weight of the polymer and more important the type of the polymer which could be used/applied for this calculation. The main problem in electrospinning is the viscosity of the solutions and polyelectrolyte effect as well as hydrogen bonding. 

Also, it is important to establish a experimental correlation between the rheology of the polymer and the possibility of fibre alignment or the formation of mats without entanglements .  For me, if a theoretical paper is not explaining the real situation is not worth for publishing.

I recommend to enrich the mathematical models with experimental results. 

Author Response

Response to Reviewer 3 Comments

The manuscript deals with mathematics of polymer chain behavior. However, it is important to mention experimental factors such as the influence of molecular weight of the polymer and more important the type of the polymer which could be used/applied for this calculation. The main problem in electrospinning is the viscosity of the solutions and polyelectrolyte effect as well as hydrogen bonding. Also, it is important to establish a experimental correlation between the rheology of the polymer and the possibility of fibre alignment or the formation of mats without entanglements.

We are grateful to the reviewer for criticism, which is very useful for our further work including experimental approaches.

We agree that the nature of the polymer chains and their molecular weight are very important in the electrospinning process. We also agree that the listed problems are actual, but they require special experimental consideration which is out of the scope of the present paper.

In our paper we studied the chains of different molecular mass which repulse each other sterically and attract by the Van der Waals forces.

The first task of this paper is to understand how the orientation of the chains affects the dynamics of the jet and formation of the fiber. In some extent, this situation is similar to so-called “mechanotropic” spinning [25, 28, 29], where due to stretching of macromolecules a solvent is squeezed out from a jet. In principle, Fig. 7 is very close to this process due to attraction forces coupling macromolecules and displacing a solvent with its further evaporation. In both cases the phase separation proceeds because of stretching and aggregation of macromolecules.      

 For me, if a theoretical paper is not explaining the real situation is not worth for publishing.

In our opinion, theoretical works should not only explain the available experiments, but also motivate researchers to set up new experiments to test the predictions of the theory. We rewrote the Conclusion and tried to formulate our main predictions more clearly. 

I recommend to enrich the mathematical models with experimental results. 

We added the references [38, 40] with a short description of some experimental results.

Reviewer 4 Report

The paper entitled ‘Orientation and aggregation of polymer chains in the straight electrospinning jet’ shows the theoretical analysis of polymer chain behavior in the jet in the electrospinning. The main concern about this study is that polymer chain length is defined by the molecular mass of the polymer. It is not considered by the author in the theoretical model. Therefore I am not sure if it is useful. It is not clear what actually the studies reveal or predict. It also lucks of any comparison and discussion related to experimental work. The authors should specify for which molecular weight the model is valid and try to verify it with any experimental results published in the literature if possible.

Author Response

Response to Reviewer 4 Comments

The paper entitled ‘Orientation and aggregation of polymer chains in the straight electrospinning jet’ shows the theoretical analysis of polymer chain behavior in the jet in the electrospinning. The main concern about this study is that polymer chain length is defined by the molecular mass of the polymer. It is not considered by the author in the theoretical model. Therefore I am not sure if it is useful. It is not clear what actually the studies reveal or predict.

We are grateful the reviewer for criticism, which made it possible to improve the paper.

The polymer chain length is proportional to its molecular weight. We formulated the main predictions of our theory in terms of molecular weight in the Conclusion.

It also lucks of any comparison and discussion related to experimental work. The authors should specify for which molecular weight the model is valid and try to verify it with any experimental results published in the literature if possible.

In this manuscript we studied the chains of different molecular weight. At the moment we found only qualitative agreements with experiment, including earlier published papers on mechanotropic spinning. We cannot exclude that the major processes governing the phase separation processes have similar nature regardless character of extension forces: electric or mechanical. Further experimental work is planned to test the predictions of the theory namely in electrospinning.

Round 2

Reviewer 3 Report

The authors change the conclusions of the work, which describes the scientific development. I agree with the publication of the article

Author Response

The authors change the conclusions of the work, which describes the scientific development. I agree with the publication of the article.

We sincerely appreciate the reviewer for a final positive assessment of our manuscript.

Reviewer 4 Report

The authors clarify the comments.

Author Response

The authors clarify the comments.

We have tried to clarify our results and believe that they will be of interest to specialists in polymer processing.